# Do All Individual Layers Help? An Empirical Study of Task-Interfering Layers in Vision-Language Models

## Abstract

Current Vision-Language Models (VLMs) have demonstrated remarkable capabilities across a wide range of multimodal tasks. Typically, in a pretrained VLM, all layers are engaged by default to make predictions on downstream tasks. Surprisingly, we find that intervening on a single layer, such as by zeroing its parameters, can improve the performance on certain tasks, indicating that some layers hinder rather than help downstream tasks. To understand when and why this occurs, we systematically investigate how individual layers influence different tasks via layer intervention (e.g., parameter zeroing). Specifically, we measure the change in performance relative to the base model after intervening on each layer and observe improvements when bypassing specific layers. This improvement can be generalizable across models and datasets, indicating the presence of **Task-Interfering Layers** that harm downstream tasks' performance. To further analyze this phenomenon, we introduce **Task-Layer Interaction Vector**, which quantifies the effect of intervening on each layer of a VLM given a task. Crucially, these task-interfering layers exhibit task-specific sensitivity patterns: tasks requiring similar capabilities show consistent response trends under layer interventions, as evidenced by the high similarity in their task-layer interaction vectors. Inspired by these findings, we propose TaLo (**Ta**sk-Adaptive **L**ayer Kn**o**ckout), a training-free, test-time adaptation method that dynamically identifies and bypasses the most interfering layer for a given task. Without any parameter updates, TaLo consistently improves performance across various models and datasets, even boosting Qwen-VL's accuracy on the Maps task in ScienceQA by up to 16.6%. Our work reveals an unexpected form of modularity in pretrained VLMs and provides a plug-and-play, training-free mechanism to unlock hidden capabilities at inference time. The source code will be publicly available.

## 1 Introduction

Vision-Language Models (VLMs) have demonstrated remarkable success across diverse domains, including medicine (Li et al., 2023b; Lin et al., 2025; Yang et al., 2025), autonomous driving (Sima et al., 2025; Guo et al., 2024), and creative industries (Wang et al., 2023; Huang et al., 2023), owing to their powerful cross-modal understanding and generation capabilities. In practical deployment, it is conventionally assumed that every layer in a VLM is actively utilized, thus justifying the use of the full model and requiring a complete computational pass during inference (Yin et al., 2024). However, our empirical investigation reveals a counterintuitive phenomenon: selectively bypassing a single layer of a pretrained model, can lead to substantial performance improvements on certain tasks. This observation naturally leads to a fundamental question: ***Do all individual layers in a pretrained VLM play a beneficial role in a specific task?***

To address the question, we first introduce layer intervention to quantify layer contributions towards a specific task: if performance on a task improves after intervening on a layer, we infer that the layer was previously hindering that task. Specially, we zero out the self-attention module of each layer, preserving residual connections while bypassing the attention mechanism, thereby nullifying the layer's learned knowledge. As shown in Figure 1, zeroing specific layers leads to substantial performance gains on particular tasks across different models, while Figure 2a provides a more sys-

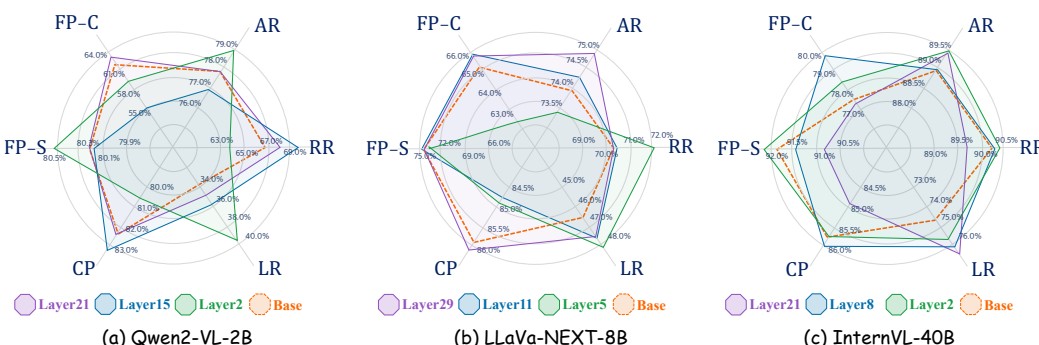

Figure 1: **Overview of the task-interfering layer phenomenon.** Each axis corresponds to a task category: AR (Attribute Reasoning), RR (Relation Reasoning), LR (Logical Reasoning), CP (Coarse Perception), FP-S (Fine-grained Perception [single-instance]), and FP-C (Fine-grained Perception [cross-instance]). Each plot shows model performance after zeroing out a single layer (solid curves), with the orange dashed line indicating the baseline performance (no intervention). In several tasks, performance improves upon layer removal, providing direct evidence for the existence of Task-Interfering Layers.

tematic and comprehensive analysis, illustrating that every task follows a characteristic pattern of performance change as different layers are intervened, reflecting task-specific sensitivities to layer functionality. We then introduce the **Task-Layer Interaction Vector** to further formalize this analysis. This vector quantifies the interaction between a task and each model layer by measuring performance changes under layer intervention. By encoding each task as such a vector, its unique sensitivity to layer manipulations becomes a computable and comparable representation. Building on this, a cluster analysis (see Figure 2b) of the correlations between these interaction vectors reveals that tasks requiring similar capabilities, such as mathematical reasoning, exhibit high similarity between their task-layer interaction vectors, indicating a strong relationship between the model's internal functional organization and the cognitive attributes of the tasks it performs, shedding light on the mechanisms behind its functional specialization.

While this phenomenon highlights the intricate functional organization within VLMs, a systematic exploration remains absent. Recent studies have noted that intervening layers (such as parameter zeroing or uniform scaling) in models can alter their general capabilities (Zhang et al., 2024; Chen et al., 2025b). However, this body of work has primarily highlighted the degradation of overall model performance, but overlooking the concurrent emergence of enhanced capabilities in certain downstream tasks. Crucially, existing research not only lacks a thorough understanding of this phenomenon but also overlooks its potential utility. Our work aims to address this critical gap. Our focus is not only on identifying what we term **Task-Interfering Layers**, which are layers whose presence actively constrains a model's potential on specific tasks, but also on uncovering the underlying mechanisms behind this phenomenon and exploring their practical applications.

Building on these motivations, we further introduce **TaLo** (**Ta**sk-Adaptive **L**ayer Kn**o**ckout), a training-free, test-time adaptive framework. TaLo dynamically selects which layers to eliminate during inference for a given task, effectively enhancing its specific capabilities. The efficacy of this approach is validated across multiple VLMs and benchmarks. For LLaVA (Li et al., 2024), applying the TaLo method yields up to a 4.7% performance gain on the Tech&Engineering task of MMMU (Yue et al., 2024). Remarkably, on the Physical Geography task of ScienceQA (Lu et al., 2022), it achieves an impressive 10.4% improvement entirely without any parameter updates or additional training, demonstrating its practical value.

We summarize our contributions as follows:

1. Through systematic layer-wise interventions, we observe that bypassing certain layers can lead to improved task performance. We refer to these as **Task-Interfering Layers**, denoting pretrained components whose pretrained knowledge are inconsistent with the objectives of specific downstream tasks.

2. We establish a quantitative framework for analyzing the relationship between tasks and model layers by introducing the **Task-Layer Interaction Vector**, enabling further examination of how similar tasks exhibit consistent responses to layer interventions.

3. We develop a practical, plug-and-play algorithm **TaLo** that leverages these insights to improve model performance at test time without any parameter updating. Using this method, LLaVA and Qwen-VL (Wang et al., 2024) achieve peak improvements of up to 10.4% and 16.6%, respectively, from 10 tasks spanning 5 benchmarks.

## 2 RELATED WORK

Our research is situated at the intersection of Model Editing, Pruning, and Test-Time Adaptation (TTA). We draw upon concepts from model editing and pruning by using parameter intervention to modulate model behavior, yet we introduce a distinct approach focused on dynamic suppression for task-specific gains. Our method, TaLo, further contributes a novel paradigm to TTA by skipping the model's layer at inference time.

### 2.1 MODEL EDITING AND PRUNING

**Model Editing** aims to modify pretrained models' behaviors or update knowledge without full retraining. Methods mainly fall into two categories. The first involves direct parameter updates, such as constrained fine-tuning to mitigate forgetting (Zhu et al., 2020) or hyper-networks for dynamic parameter adjustment (Cao et al., 2021). However, these are challenging to scale to large language models due to their parameter size. MEND (Mitchell et al., 2022a) addresses this by using low-rank gradient decomposition for efficient updates. The second category, Locate-and-Edit, identifies key parameters (e.g.,"knowledge neurons") and applies targeted modifications (Meng et al., 2023a; Dai et al., 2022; Meng et al., 2023b). While enhancing interpretability, this approach is often labor-intensive and limited in scalability. Some methods instead maintain original parameters and use auxiliary modules for editing (Mitchell et al., 2022b). In multimodal settings, editing Vision-Language Models (VLMs) requires unique strategies. Directly porting LLM methods is ineffective; instead, recent work (Chen et al., 2025a) proposes manipulating intermediate visual representations by identifying and editing regions most relevant to the target prompt, minimizing interference with unrelated features while preserving efficiency.

**Model Pruning** is distinct from model editing. It is a technique for model compression and acceleration. It operates by removing redundant or unimportant components, such as weights, neurons, or entire layers, to reduce the model's size and improve inference speed, while aiming to preserve original performance (Cheng et al., 2024; Ma et al., 2020; He et al., 2017; Dumitru et al., 2024; Muralidharan et al., 2024; Siddiqui et al., 2024; Yin et al., 2023). Its primary objective is efficiency optimization rather than enhancing or correcting the model's knowledge capabilities.

While our method, TaLo, shares the objective of enhancing performance with model editing and pruning, its strategy is fundamentally different. Unlike model editing, which permanently injects new knowledge, or pruning, which permanently removes components for compression, the proposed TaLo temporarily suppresses harmful reasoning paths through reversible, test-time interventions. This demonstrates that performance can be improved not only by adding or removing components, but also by strategically inhibiting existing ones during inference.

### 2.2 TEST-TIME ADAPTATION

Test-Time Adaptation (TTA) aims to dynamically adjust models to shifting data distributions during inference, a crucial step for robust deployment in real-world scenarios. Prevailing approaches either update model components like weights or normalization statistics using test batches (Iwasawa & Matsuo, 2021; Wang et al., 2021; Yi et al., 2023; Sun et al., 2020; Schneider et al., 2020), or, particularly for vision-language models, fine-tune learnable prompts (Zhang et al., 2022; Shu et al., 2022; Feng et al., 2023). Other methods (Karmanov et al., 2024) perform zero-shot classification using test-time feature caching. Our work introduces a distinct layer-intervention approach to TTA. Guided by a few test-time samples, our method identifies and dynamically zeroes out task-interfering layers during inference. This training-free strategy avoids the large-scale param-

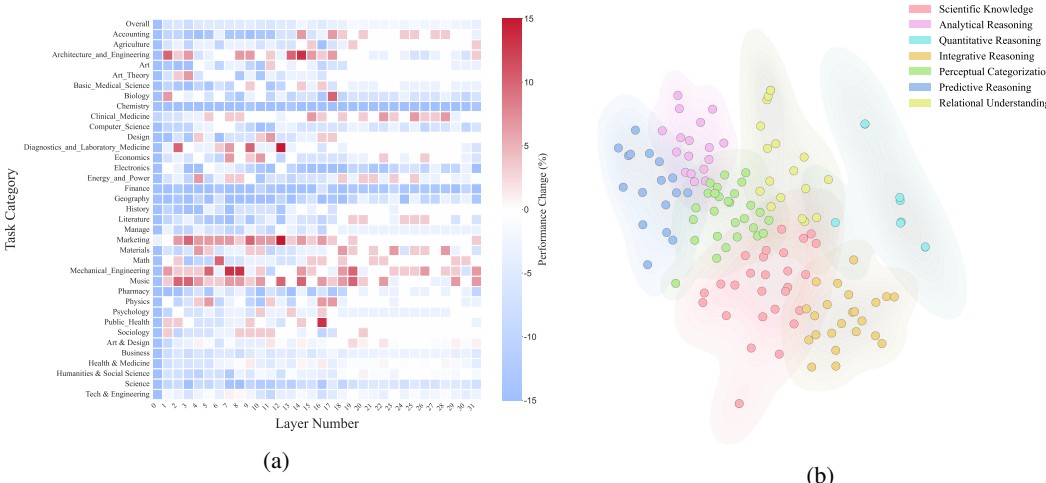

(a)

(b)

Figure 2: **Empirical Validation of the Task-Interfering Layers.** **(a)** Visualization of the percentage change in accuracy across tasks after zeroing each layer on *LLaVA-Next-LLaMA3-8B*. Red indicates performance improvements relative to the base model, while blue indicates degradation. Many tasks show performance gains under layer interventions, indicating that interfering layers are commonly exist in VLMs. **(b)** The t-SNE visualization of task clusters. Each point in the figure represents a task encoded as a Task-Layer Interaction Vector $\mathbf{v}^{(\mathcal{T})}$. Tasks are clustered based on their pairwise similarities measured by Pearson correlation $\rho_{ij}$ between different vectors, with tasks requiring similar capabilities forming coherent clusters. The color-coded clusters correspond to different types of tasks, indicating that tasks with shared cognitive demands exhibit similar intervention responses, reflecting a structured functional layout in LLaVA (Li et al., 2024) (See Table 11 for complete clustering details).

eter updates and complex prompt modifications inherent in prior methods. Consequently, it enables efficient, plug-and-play adaptation with minimal computational overhead, as the original model parameters remain intact and reusable across tasks.

## 3 DISCOVERING AND CHARACTERIZING TASK-INTERFERING LAYERS

We begin by using parameter intervention to uncover the commonly existing phenomenon of Task-Interfering Layers. Building on this, we introduce the Task-Layer Interaction Vector, which enables a more systematic analysis, effectively uncovering patterns and revealing the coherence of layer-level interference.

### 3.1 DISCOVERING TASK-INTERFERING LAYERS

Our aim is to isolate and quantify the contribution of individual layers to specific task capabilities. To achieve this, we employ parameter intervention to systematically probe the functional role of each layer inspired by Zhang et al. (2024); Chen et al. (2025b). Specifically, for each layer, we replace the parameters of the self-attention module with zeros or a uniform distribution (i.e., setting every parameter to an identical value $1/N$, where $N$ corresponds to the matrix's first dimension) and evaluate the resulting change in task accuracy against the unmodified model. For parameter zeroing, it effectively nullifies the attention mechanism, leaving only the residual connection, which facilitates direct communication between distant layers, bypassing intermediate transformations. As for Uniform Scaling, it reduces the complex attention operation to a simple global averaging of the input features, causing the output to become a rank-one matrix. Our hypothesis is that a performance increase after intervention suggests that the layer was hindering task performance, indicating its role as a Task-Interfering Layer. Conversely, a performance drop indicates that the layer contributed positively to the task.

We then apply parameter zeroing intervention to LLaVA-Next (Li et al., 2024), a model consisting of 32 layers, and evaluate performance on the MMMU (Yue et al., 2024) dataset. As shown in Figure 2a, 54.1% of tasks exhibit performance gains exceeding 5% when a single layer's parameters are zeroed. Similar trends are observed on other models and datasets: for Qwen-VL, the proportion reaches 75.6% (shown in Figure 18b). To further validate the generality of our findings, we present additional experiments on diverse models, benchmarks, and intervention strategies in Appendix C.3. This consistent pattern provides direct empirical evidence for the existence of Task-Interfering Layers whose activation hinders rather than helps task performance.

## 3.2 Characterizing Task-Interfering Layers

**Modeling Task-Layer Interaction into Vector Space.** To uncover the systematic response patterns of tasks to layer interventions, we introduce the Task-Layer Interaction Vector. This vector is designed to model the relationship between a task's performance and interventions applied to each layer of the model.

Specifically, the Task-Layer Interaction Vector is a representation that characterizes a task's sensitivity to each model layer. Each dimension of the vector corresponds to a network layer and captures the change in task accuracy caused by intervening on that layer relative to the base model. A positive value indicates that the layer interferes with the task, which manifests as an improvement in accuracy upon intervention. Conversely, a negative value indicates that the layer contributes positively to the task, reflected in a drop in accuracy when the layer is modified. More formally, for task $\mathcal{T}$, and a model with $L$ layers, the **Task-Layer Interaction Vector** is defined as:

$$\mathbf{v}^{(\mathcal{T})} = \left( v_1^{(\mathcal{T})}, v_2^{(\mathcal{T})}, \ldots, v_L^{(\mathcal{T})} \right) \in \mathbb{R}^L, \tag{1}$$

where each element $v_i^{(\mathcal{T})}$, referred to as the **layer sensitivity score**, quantifies the change in task performance upon intervention at layer $i$. Formally, it is defined as

$$v_i^{(\mathcal{T})} = \text{Acc}\left( \mathcal{M}_{\text{intv}}^{(i)}, \mathcal{T} \right) - \text{Acc}\left( \mathcal{M}_{\text{base}}, \mathcal{T} \right). \tag{2}$$

Here, $\text{Acc}(\cdot, \mathcal{T})$ denotes the accuracy on task $\mathcal{T}$, $\mathcal{M}_{\text{base}}$ is the base model without intervention, and $\mathcal{M}_{\text{intv}}^{(i)}$ is the model with the $i$-th layer's parameters intervened.

Through this vector representation, we abstract the influence of each layer on a task into a structured representation within a high-dimensional vector space. This provides a quantifiable and comparable analytical tool for subsequent pattern analysis.

**Characterizing Task-Layer Interaction Patterns.** We assume that tasks drawing upon the same underlying cognitive skills (e.g., numerical reasoning or arithmetic reasoning) should engage similar internal processing pathways within the model. Since Task-Layer Interaction Vector $\mathbf{v}^{(\mathcal{T})}$ captures a task's dependence on each layer, reflecting how it propagates through the model's architecture. We hypothesize that related tasks will exhibit highly correlated interaction vectors.

To validate this hypothesis, we conduct a systematic analysis across 6 benchmarks and nearly 100 tasks. For each pair of tasks, $\mathcal{T}_i$ and $\mathcal{T}_j$ (hereafter, we use indices $i$ and $j$ as shorthand for the corresponding tasks in this section), we compute their Pearson correlation coefficient, $\rho_{ij} = \text{Corr}(\mathbf{v}^{(i)}, \mathbf{v}^{(j)})$, and define a distance metric as $d_{ij} = 1 - \rho_{ij}$. This ensures that more similar tasks have a smaller distance, providing a solid basis for clustering.

As shown in Figure 2b, the results confirm our hypothesis: tasks that rely on shared abilities cluster together in the task-layer interaction space, reflecting their common internal processing mechanisms. This suggests that tasks sharing underlying cognitive or domain-specific demands exhibit highly similar sensitivity patterns to layer interventions, revealing a structured organization of functional dependencies across the model's architecture. For instance, one prominent cluster is dominated by **quantitative reasoning** tasks (e.g., *numeric commonsense*, *arithmetic reasoning*, and *geometry*). Another distinct cluster groups together domain-specific **scientific tasks** such as *Physics*, *Chemistry*, and *Scientific Reasoning*, reflecting their shared reliance on formal scientific knowledge. This demonstrates a strong alignment between the model's internal response to layer interventions

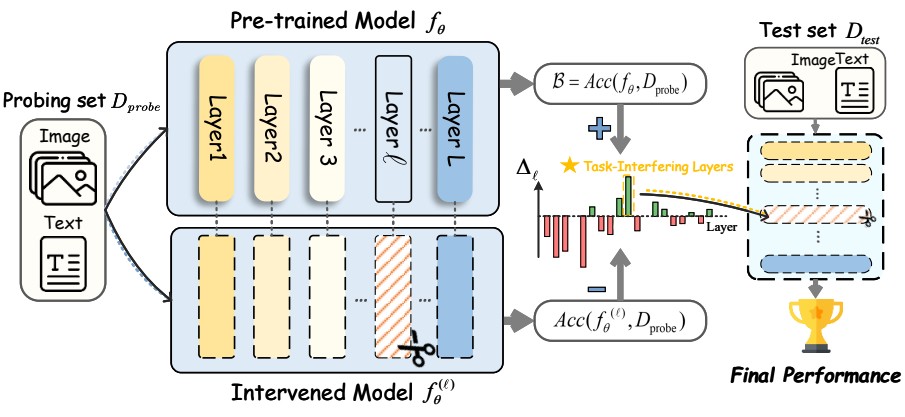

**Stage1: Dynamic Layer Selection**          **Stage2: Layer Knockout**

Figure 3: **Framework of TaLo.** TaLo first dynamically selects the Task-Interfering layer for a specific task and knocks out that layer in the final evaluation procedure.

and the underlying cognitive structure of tasks, revealing that task-interfering layers generalize across tasks with similar abilities. This generalization suggests that layer sensitivity is determined by functional demands, not task-specific properties, enabling reliable estimation of interfering layers from a few representative samples.

**Consistency across Intervention Methods.** To validate that the Task-Interfering Layer phenomenon is not a methodological artifact, we examine the consistency of findings across two distinct intervention strategies. For each task $\mathcal{T}$ and layer $i$, we measure the model's accuracy under both intervention types, yielding two performance scores: $\text{Acc}(\mathcal{M}_{\text{zero}}^{(i)}, \mathcal{T})$ and $\text{Acc}(\mathcal{M}_{\text{unif}}^{(i)}, \mathcal{T})$. We perform this analysis across six benchmarks. To visualize the relationship, we generate scatter plots (Figure 4) where each point represents a single task-layer pair, with its coordinates determined by the accuracies under uniform scaling (x-axis) and parameter zeroing (y-axis). To quantify the level of agreement, we then compute the Pearson correlation coefficient across all points for each benchmark. The results reveal a strong and statistically significant positive correlation across all benchmarks. This reproducibility across fundamentally different intervention strategies strongly reinforces the validity of our findings. It affirms that the existence of Task-Interfering Layers is not an artifact of a specific intervention choice, but rather reflects an intrinsic property of the model arising from task conflicts during pretraining.

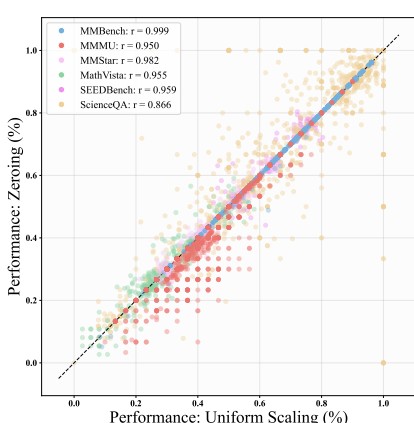

Figure 4: Consistency analysis of different interventions.

## 4 TASK-ADAPTIVE LAYER KNOCKOUT

Building upon the discovery of Task-Interfering Layers in VLMs, and inspired by advances in dynamic model adaptation (Wang et al., 2018; Zhao et al., 2025; Cao et al., 2024; Sun et al., 2025a; Cao et al., 2023), we propose a simple yet effective algorithm—**Ta**sk-Adaptive **L**ayer Kn**o**ckout **(TaLo)**-a training-free method for task-level model customization at test time. TaLo operates in two stages: (1) dynamic layer selection for a specific task, and (2) task-interfering layer knockout on the model. This approach enables performance enhancement on specific tasks by strategically exploiting the task-interfering layers within the model.

Our approach is guided by the principle of targeted intervention. The objective is to identify an optimal layer whose modification unlocks a model's latent, task-specific abilities. The search begins by

Table 1: The performance of **TaLo** applied to two models (Zeroing): *LLaVA-Next-LLaMA3-8B*, *Qwen2-VL-2B* across five benchmarks. MMStar (*Logical reasoning* annotated as L-R), MM-Bench (*Physical property reasoning* annotated as PP-R, *image emotion* annotated as I-E), MMMU (*Tech&Engineering* annotated as T&E, *Health&Medicine* annotated as H&M), ScienceQA (*Physical Geography* annotated as P-G), and SEEDBench (*Visual Reasoning* annotated as V-R, *Text Understanding* annotated as T-U). Additional results for TaLo on various tasks are provided in Table 9 and Table 10. Shot number indicates probe set size. Base performance may vary across tasks, especially those with limited samples under different shot settings. To ensure a fair evaluation, we therefore focus primarily on the relative improvement over the base model.

| Model | Shots | MMStar | | MMBench | | MMMU | | ScienceQA | | SEEDBench | | Avg |
|---|---|---|---|---|---|---|---|---|---|---|---|---|
| | | Math | I-R | PP-R | I-E | T&E | H&M | P-G | Maps | V-R | T-U | |
| LLaVA | 10 shots | 32.9 ↑2.5 | 58.3 ↑4.2 | 55.3 ↑7.8 | 65.6 ↑0.6 | 35.2 ↑1.1 | 42.2 ↑5.1 | 34.5 ↑6.9 | 16.7 ↑2.4 | 70.4 ↑1.3 | 53.6 ↑7.2 | 3.91↑ |
| | 15 shots | 33.2 ↑3.0 | 57.9 ↑4.3 | 56.0 ↑3.8 | 69.3 ↑0.7 | 37.5 ↑8.3 | 35.3 ↑5.9 | **41.4** ↑10.4 | 23.8 ↑7.1 | 70.8 ↑1.1 | 55.4 ↑9.0 | 5.36↑ |
| | 20 shots | 33.9 ↑3.0 | 58.7 ↑4.3 | 53.4↑4.8 | 66.4↓1.5 | 29.5 ↓3.6 | 46.2−0.0 | 34.5 ↑3.5 | 26.2 ↑9.5 | 72.5 ↑2.0 | 55.4 ↑5.4 | 2.74↑ |
| Qwen-VL | 10 shots | 44.8 −0.0 | 54.8 ↓1.9 | 48.6 ↑8.5 | 69.4 ↑1.3 | 28.6 ↑0.6 | 34.5 ↓1.6 | 31.0−0.0 | 31.0 ↑2.4 | 70.8 ↓1.5 | 57.1 ↑8.9 | 1.67↑ |
| | 15 shots | 44.2 ↑0.5 | 54.2 ↓1.6 | 55.3 ↑1.2 | 65.7 ↓0.7 | 24.4 ↓1.9 | 35.8 ↓2.9 | 34.5 ↑3.5 | **45.2** ↑16.6 | 71.2 ↓2.6 | 55.4 ↑1.8 | 1.39↑ |
| | 20 shots | 38.2 ↓1.2 | 55.3 ↓0.5 | 60.3 ↑4.8 | 67.9 ↑0.7 | 26.1 ↓0.5 | 23.5 ↓5.9 | 37.9−0.0 | 16.7−0.0 | 71.3↓0.8 | 44.6 −0.0 | 0.34↓ |

establishing a performance baseline. For an $L$-layer model $f_\theta$ with parameters $\theta = \{\theta_1, ..., \theta_L\}$ and a probing set $D_{\text{probe}} = \{(x_i, y_i)\}_{i=1}^N$ sampled from a given downstream task in an $N$-shot setting, we define the baseline score $\mathcal{B}$ on the unmodified model: $\mathcal{B} = Acc(f_\theta, D_{\text{probe}})$. In our experiments, $Acc$ is accuracy on probing set $D_{\text{probe}}$. Then, we systematically test each layer's potential. For each layer $\ell$, we apply an intervention $I$ (Here, we utilize parameter zeroing, $I(\theta_\ell) = 0$) to create a modified model $f_\theta^{(\ell)}$ and measure the resulting accuracy gain $\Delta_\ell$:

$$\Delta_\ell = Acc(f_\theta^{(\ell)}, D_{\text{probe}}) - Acc(f_\theta, D_{\text{probe}}). \tag{3}$$

This iterative process reveals how each layer's function contributes to the specific task. The search concludes when we identify the optimal layer $\ell^*$ responsible for the maximal positive performance gain:

$$\ell^* = \underset{\ell \in \{1, ..., L\}}{\arg\max} \ \Delta_\ell. \tag{4}$$

This layer is then selected as the task-interfering layer. If no such layer exhibits a statistically significant sensitivity peak, we retain the original model without any modification. Having identified the layer, we proceed to apply the intervention (i.e., knockout) to this layer during inference. We then evaluate the intervened model on the held-out test samples of the target task, which are those not included in the initial probe set, to ensure an unbiased assessment.

## 5 EXPERIMENTS

### 5.1 SETUPS

**Models and Benchmarks.** We conducted experiments on three VLMs of varying scales: Qwen2-VL-2B (Wang et al., 2024), LLaVA-Next-LLaMA3-8B, and InternVL2-26B (Chen et al., 2024b). To assess the impact of our interventions, we evaluated both the intervened and original models across five key multiple-choice question (MCQ) benchmarks: MMStar (Chen et al., 2024a), MMBench (Liu et al., 2024), MMMU (Yue et al., 2024), ScienceQA (Lu et al., 2022), and SEED-Bench (Li et al., 2023a). Further details on the benchmarks and model configurations are provided in Appendix A.

**Baselines.** We aim to achieve task-specific gains via training-free, plug-and-play layer intervention. All comparisons are against the original pretrained model. In Section 5.3, we also evaluate fine-tuning methods using the same few-shot samples as TaLo.

**Implementation Details.** All experiments were conducted on a single *80GB A100 GPU* under identical environments, ensuring reproducibility and fair comparisons. Evaluations used the standardized VLMEvalKit framework (Duan et al., 2024). For more experimental details, please refer to the Appendix B.

Table 2: The performance of **TaLo** applied to *InternVL2-26B* (Zeroing) across multiple tasks. *Persuasive strategies* annotated as P-S, *Basic economic principles* annotated as B-EP, *Physical Geography* annotated as P-G, *Geography* annotated as Geo, The Americas: geography annotated as A:Geo, and *G2T* annotated as Genes to traits.

| Model | Shots | Tasks | | | | | | | | Avg |
|---|---|---|---|---|---|---|---|---|---|---|
| | | P-S | B-EP | P-G | Solutions | Geo | A:Geo | G2T | Materials | |
| InternVL | **10 shots** | 58.3 ↑8.3 | 27.9 −0.0 | 34.5 −0.0 | 40.0 ↑6.7 | 35.4 ↑2.1 | **35.0** ↑10.0 | 25.0 ↓3.1 | 38.5 ↑1.3 | 3.16↑ |
| | **15 shots** | 33.3 ↑8.3 | 32.6 ↑2.4 | 31.0 ↑6.9 | 31.1 −0.0 | 27.1 −0.0 | 20.0 −0.0 | 37.5 ↑3.1 | 41.0 ↓1.3 | 2.43↑ |

Table 3: Comparison of TaLo with other methods. Each entry reports performance and running time ($\times 10^2$ s). TaLo attains higher accuracy with lower adaptation time.

| | Math | | | | | | | | Instance Reasoning | | | | | | | |
|---|---|---|---|---|---|---|---|---|---|---|---|---|---|---|---|---|
| | Base | Merge | LoRA-FT | | OFT | | TaLo | | | Base | Merge | LoRA-FT | | OFT | | TaLo | |
| Shots | Score↑ | Score↑ | Score↑ | Time↓ | Score↑ | Time↓ | Score↑ | Time↓ | Shots | Score↑ | Score↑ | Score↑ | Time↓ | Score↑ | Time↓ | Score↑ | Time↓ |
| 10shots | 30.42 | 32.38 | 31.25 | 1.75 | 30.83 | 14.67 | **32.92** | 1.70 | 10 shots | 54.16 | 53.33 | 55.83 | 2.06 | 55.42 | 15.34 | **58.33** | 1.33 |
| 15shots | 30.21 | 32.63 | 32.17 | 2.53 | 32.34 | 21.91 | **33.19** | 1.76 | 15 shots | 53.61 | 52.34 | 55.32 | 2.87 | 56.17 | 25.43 | **57.87** | 1.81 |
| 20shots | 30.87 | 31.76 | 33.04 | 3.46 | 33.04 | 29.16 | **33.91** | 2.84 | 20 shots | 54.35 | 53.04 | 55.65 | 4.06 | 56.52 | 33.81 | **58.69** | 2.89 |
| Avg | 30.50 | 32.26 | 32.15 | 2.58 | 32.07 | 21.91 | **33.34** | 2.10 | Avg | 54.04 | 52.90 | 55.60 | 2.99 | 56.04 | 24.86 | **58.30** | 2.01 |

## 5.2 EXPERIMENTAL RESULTS

The results presented in Tables 1 and 2 compellingly demonstrate the effectiveness of TaLo. Specifically, for the LLaVA model, TaLo achieves a peak performance gain of **10.4%** across five benchmarks and ten different tasks. Notably, its average performance is consistently higher than the baseline model across all three shot settings. Similarly, on the same set of tasks, the Qwen-VL model shows a maximum performance increase of **16.6%**. For InternVL, as shown in Table 2, while our evaluation was conducted on a smaller set of eight tasks, the findings are consistent: TaLo delivers an average performance improvement in all configurations, with a peak gain reaching **10.0%** (see Appendix C.2 for more results and analysis).

## 5.3 MORE ANALYSES

In this subsection, we provide a streamlined yet comprehensive analysis of our method using MM-Star (Chen et al., 2024a), which is a consolidated benchmark spanning six diverse VLM task categories, to validate effectiveness and uncover key patterns systematically.

**Comparison Study.** To further evaluate TaLo, we compare it with model merging (Chen et al., 2025b; Yang et al., 2024; Ilharco et al., 2023) as well as fine-tuning methods (LoRA (Hu et al., 2021), OFT (Qiu et al., 2024)), all aiming to improve task-specific performance. For all fine-tuning experiments, we select the checkpoint from the epoch that achieves the best validation performance upon convergence. We use LLaVA-Next-LLaMA3-8B as the base VLM, and the same few-shot samples for both TaLo and the fine-tuning approaches. Detailed configurations are provided in Appendix B.

As shown in Table 3, TaLo achieves superior performance compared to both merging and fine-tuning baselines in less time across most settings. Crucially, unlike any of the baselines, TaLo requires neither external models nor any form of training or task-specific parameter updates. Instead, it enables on-the-fly adaptation through minimal, dynamic intervention within the base model, highlighting its efficiency, simplicity, and strong practical applicability.

**Extending TaLo to Multi-Layer Interventions.** To further explore the potential of TaLo beyond single-layer intervention, we extend the method to jointly intervene on multiple layers. Specifically, we enhance the original TaLo procedure by performing an iterative search over layer pairs: after identifying the most beneficial single layer for a given task, we systematically apply a second intervention to each of the remaining layers and measure the resulting change in performance. The optimal two-layer combination is selected as the pair that yields the maximum performance gain relative to the best single-layer intervention.

Table 4: Results of TaLo on *LLaVA* under two-layer intervention (10-shot). '✗' marks cases where a second Task-Interfering Layer could not be identified. Details of the MMStar are provided in Appendix A.

| Metric | MMStar | | | | | |
|---|---|---|---|---|---|---|
| | CP | FP | IR | S&T | LR | Math |
| **Task-Interfering Layer** | L1, L6 | L15, L29 | L11, ✗ | L31, ✗ | L1, L8 | L6, L13 |
| **Performance (two layers)** | 61.9 ↓3.8 | 40.0 ↓0.5 | 56.2 ↑1.0 | 31.0 ↑0.5 | 37.1 ↓3.8 | 25.7 ↓3.8 |
| **Performance (single layer)** | 63.8 ↓2.9 | 41.9 ↑0.9 | 57.6 ↑3.8 | 38.6 ↓3.8 | 31.0 ↓2.9 | 32.9 ↑2.5 |

Due to the rapid growth in computational cost when exploring multi-layer interventions, we conduct our analysis using two-layer combinations as a representative sample. Despite this simplification, our findings remain highly informative. As shown in Table 4, for several tasks, no second task-interfering layer can be identified, suggesting task-interfering layers may be sparse and strong inter-layer interactions further obscure individual roles, thereby justifying TaLo's focus on single-layer interventions.

# 6 DISCUSSIONS

**Limitation and Future Work.** Our analysis relies on existing benchmarks, and the predefined task categories within them may influence the layer sensitivity patterns we observed. Future work could therefore validate our findings across more granular task decompositions, which may help establish the generality of this phenomenon. With respect to our method, TaLo, we acknowledge that it is a minimalist framework. Our primary objective was not to achieve state-of-the-art performance, but rather to provide a simple, plug-and-play solution that serves as a proof-of-concept for the practical utility of the Task-Interfering Layer phenomenon. There are several promising avenues to explore, such as developing more sophisticated dynamic layer selection mechanisms, investigating better multi-layer modulation strategies, or even incorporating adaptive sampling techniques (Cao & Tsang, 2021; Cao et al., 2023). We are optimistic that future research can extend TaLo's capabilities, applying its principles to a broader and more complex range of scenarios.

**Hypothesis and Explanation.** We further offer a hypothesis to explain why certain layers may become task-interfering. Modern large models are pretrained on diverse and multi-task data, where each layer learns a compromise representation, which approximates a global optimum across all tasks. However, this global optimum may deviate from the local optimum for any specific task. We conjecture that Task-Interfering Layers capture features that, while beneficial on average, introduce noise or misalignment when applied to a particular task. By zeroing out or uniformly scaling these layers, TaLo effectively suppresses or rebalances their influence, which may prevent the propagation of task-irrelevant or even detrimental information. This intervention, we hypothesize, steers the model's internal computation toward a more favorable region in the parameter space that better aligns with the target task's local optimum, thereby effectively improving performance without any parameter updates.

# 7 CONCLUSION

Through extensive empirical analysis, we reveal the existence of specific layers within pretrained Vision-Language Models (VLMs) that actively suppress performance on certain downstream tasks. We term these Task-Interfering Layers, as strategically bypassing them yields significant performance improvements. Our further investigation uncovers a crucial pattern: tasks that demand similar functional abilities exhibit highly consistent response patterns to layer interventions. This suggests that the interference phenomenon is not random but is systematically organized around the model's functional capabilities, allowing the effects of Task-Interfering Layers to generalize across related tasks. Based on these findings, we introduce TaLo, a training-free adaptation method that identifies and bypasses these interfering layers at inference time. The strong performance of TaLo across a diverse range of models demonstrates that simple, targeted architectural intervention can be a powerful and efficient strategy for model adaptation, obviating the need for any parameter updates.

## REPRODUCIBILITY STATEMENT

The relevant code is provided in the supplementary materials. Details of the experimental benchmarks and models are listed in Appendix A, while a comprehensive description of the experimental setup for the mechanism studies can be found in Appendix B.

## ETHICS STATEMENT

Our research on Task-Interfering Layers in Vision-Language Models involves no human subjects or sensitive data. We uphold transparency, accuracy, and fairness, and commit to mitigating biases in AI. Code will be released to promote open and responsible research.

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

## THE USE OF LARGE LANGUAGE MODELS

We declare that large language models (LLMs) were employed to assist with the refinement of this manuscript, specifically, for grammar checking, language polishing, and improving the clarity and fluency of the text. Additionally, LLMs were used in a limited capacity for minor debugging and syntactic correction of code snippets included in the work.

## A MODELS AND BENCHMARKS

We present all the models used in our experiments in Table 5, and list all the benchmarks we utilize in Table 6.

| Name | Size | Huggingface ckpt |
|---|---|---|
| LLaVA-Next-LLaMA3 (Li et al., 2024) | 8B | `llava-hf/llama3-llava-next-8b-hf` |
| Qwen2-VL (Wang et al., 2024) | 2B | `Qwen/Qwen2-VL-2B-Instruct` |
| InternVL2 (Chen et al., 2024b) | 26B & 40B | `OpenGVLab/InternVL2-40B` |

Table 5: Details of the models used in our experiments.

| Benchmark | Category | Huggingface URL |
|---|---|---|
| MMStar (Chen et al., 2024a) | MCQ | `Lin-Chen/MMStar` |
| MMBench-EN (Liu et al., 2024) | MCQ | `lmms-lab/MMBench` |
| MMMU-VAL (Yue et al., 2024) | MCQ | `MMMU/MMMU` |
| ScienceQA-VAL (Lu et al., 2022) | MCQ | `derek-thomas/ScienceQA` |
| MathVista-MINI (Lu et al., 2024) | VQA | `AI4Math/MathVista` |
| SEEDBench-IMG (Li et al., 2023a) | MCQ | `lmms-lab/SEED-Bench` |

Table 6: Details of the benchmarks used in our experiments.

In this work, all datasets are evaluated using accuracy as the sole metric. The majority of datasets: MMStar, MMMU, SEEDBench, MMBench, and ScienceQA are multiple-choice question (MCQ) benchmarks, where the model's predicted option is extracted from its output and matched against the ground truth. MathVista, while formulated as a vision-question-answering (VQA) task, also employs direct string matching between generated responses and reference answers in its official evaluation, ensuring consistency in the metric across all tasks.

Specifically, MMStar is a comprehensive benchmark with 250 balanced samples across six core capabilities: Coarse Perception, Fine-grained Perception, Instance Reasoning, Logical Reasoning, Math, and Science & Technology. MMBench contains 2,974 MCQs assessing a wide range of abilities, including Coarse Perception, Fine-grained Perception (both single and cross instance), Instance Reasoning, Logic Reasoning, Attribute Reasoning, and Relation Reasoning. MMMU spans 30 disciplines, including Art & Design, Business, Science, Health & Medicine, Humanities & Social Sciences, and Engineering, covering 183 subfields with 30 types of heterogeneous images (e.g., charts, diagrams, maps, tables, musical scores, chemical structures), focusing on advanced perception and reasoning with domain-specific knowledge. SEEDBench comprises 19,000 human-annotated MCQs, covering 12 evaluation dimensions, including image understanding. MathVista is a challenging benchmark that combines diverse mathematical and visual reasoning tasks, consisting of 6,141 examples drawn from 28 existing multimodal math-related datasets and three newly curated datasets. Finally, ScienceQA consists of 21,208 multimodal science questions collected from elementary and high school curricula. This diverse and rigorous selection of benchmarks enables a comprehensive evaluation of task-specific abilities under a unified accuracy metric.

## B  Experimental Details

In our TaLo experiments, the procedure for identifying the optimal intervention layer begins with task definition and sample preparation. We first identify the target task according to the dataset's metadata, after which we draw samples from a probing pool held entirely separate from the final test set to prevent any data overlap.

Our identification process follows an iterative pipeline. We first establish a baseline performance by evaluating the unmodified model on an initial set of probing samples. If the baseline accuracy reaches 100%, the sample set is considered uninformative and is discarded; a new set is then drawn from the probing pool, and the baseline is re-evaluated. Once the baseline is established, we proceed with a systematic, layer-by-layer parameter intervention and measure the performance gain for each. If a unique layer yields the maximum positive gain, it is designated as the optimal target. In cases where multiple layers tie for the best performance or no layer produces a positive gain, we initiate a multi-round, augmented sampling strategy to resolve the ambiguity. This involves supplementing the set with an additional $shot/2$ samples for re-evaluation, followed by a further $shot/4$ samples if the tie persists. Should a unique optimal layer still not be identified after these two rounds, we select the layer with the highest index among the final candidates to ensure robustness(Yin et al., 2023; Gromov et al., 2024; Sun et al., 2025b; Men et al., 2024).

For our fine-tuning experiments, we employ two parameter-efficient fine-tuning (PEFT) methods: LoRA (Low-Rank Adaptation) and its variant OFT (Orthogonal Finetuning). All experiments are conducted on the LLaVA-Next-8B model. In the case of LoRA, we set the rank $r = 8$ and scaling factor $\alpha = 16$, and apply the adapter modules to all linear projections in both the language and vision pathways. This full-architecture adaptation strategy ensures comprehensive alignment of both visual and textual representations during fine-tuning. To ensure that the model truly understands the knowledge underlying the questions during fine-tuning, rather than simply memorizing the options, we format the answers as "option + option content". This approach helps the model learn the specific meaning of each option and its relationship to the question.

For model merging experiments, the LLM used for merging is DeepSeek-R1-Distill-Llama-8B(DeepSeek-AI, 2025), with a fusion coefficient $\lambda$ of 0.9. However, as shown in Table 3, the effectiveness of model merging is highly sensitive to both the choice of the external LLM and the target task, suggesting that its performance is not robust across configurations and requires careful, task-specific tuning.

## C  Additional Experimental Results and Analysis

### C.1  Specific clustering details

Table 11 provides the comprehensive list of tasks included in each of the seven clusters. From the table, the clustering appears to meaningfully group tasks by functional similarity. For instance, Cluster 1 brings together `numeric commonsense`, `arithmetic reasoning`, and `math word problem`—all clearly centered on numerical understanding and calculation. This suggests the method successfully identifies and isolates quantitative reasoning as a coherent capability.

Similarly, Cluster 3 stands out by grouping domain-specific scientific tasks—`Astronomy`, `Chemistry`, and `Scientific Reasoning`—into a unified theme, reflecting shared reliance on formal scientific knowledge.

### C.2  Additional Analysis of TaLo

**Ablation study of the Intervention Method**   While earlier results (Section 3.2) suggest that different intervention types induce similar layer-wise trends overall, their impact can vary subtly across individual tasks. To examine these influences in a controlled yet representative setting, we turn to MMStar and MMBench: two balanced benchmarks that aggregate data from multiple sources, offering moderate task diversity and comprehensive coverage. It avoids the extremes of highly specialized or overly narrow datasets, making it well-suited for studying how these factors interact with TaLo's behavior.

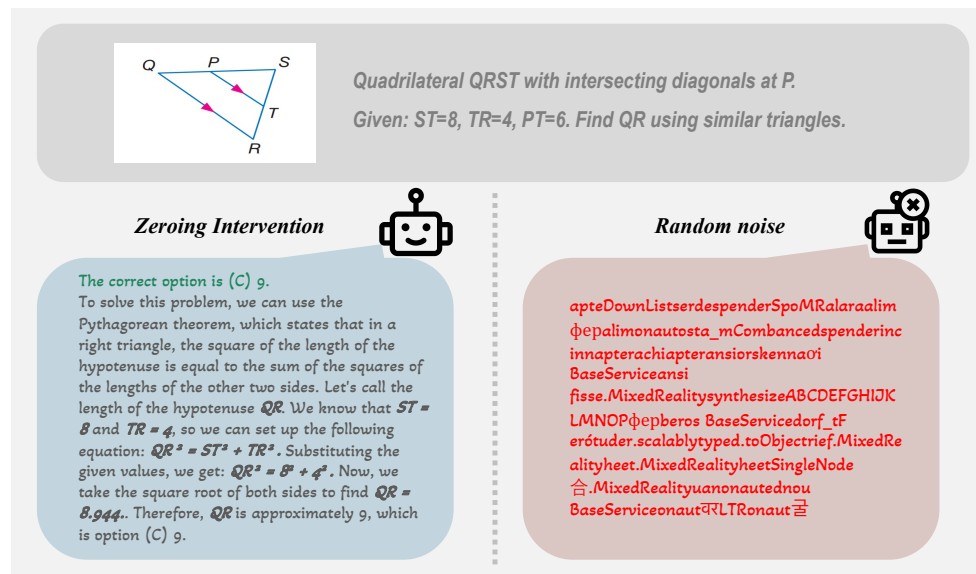

Figure 5: Qualitative case study on the random noise intervention method.

In addition to parameter zeroing and uniform scaling, we also explored replacing layer weights with their mean value and random noise. However, experiments show that injecting random noise severely destabilizes the model, effectively erasing the learned representations (as shown in Figure 5). The outputs become incoherent, often degenerate, with little connection to the input. This suggests that the pre-trained parameters, even when suboptimal for a specific task, still encode essential structural and semantic priors critical to model functionality.

Given this breakdown in basic competence, we focus our analysis on milder, more controlled interventions: zeroing, uniform scaling, and mean replacement, which preserve the model's foundational knowledge while allowing targeted modulation. These methods strike a better balance between perturbation and stability, enabling meaningful study of layer-wise task adaptation without collapsing overall performance.

| Intervention | MMStar | | | | | | Avg |
|---|---|---|---|---|---|---|---|
| | Coarse perception | Fine-grained perception | Instance reasoning | Science & technology | Logical reasoning | Math | |
| Zeroing | 63.8 ↓2.9 | 41.9 ↑0.9 | 57.6 ↑3.8 | 32.9 ↑1.0 | 38.6 ↓3.8 | 31.0 ↑2.9 | 0.32↑ |
| Uniform scaling | 66.2 ↑2.4 | 41.0 ↓0.4 | 51.0 ↓2.8 | 24.8 ↓5.2 | 40.0 ↓1.0 | 34.3 ↑3.8 | 0.53↓ |
| Mean replacement | ✗ | 34.8 ↓6.2 | 52.9 ↓0.9 | 27.6 ↓3.4 | 41.9 ↑1.4 | 30.5 ↑1.0 | 1.14↓ |

Table 7: Results of TaLo on *LLaVA-Next-Llama3-8B* under different intervention methods. The last column (**Avg**) reports the mean change across tasks. ✗ indicates the method failed to find the Task-Interfering layer.

To further examine the impact of three intervention types, we conduct ablation studies across a wide range of tasks on two benchmark datasets, using a consistent 10-shot setting. As shown in Table 7,8, we observe that zeroing and uniform scaling yield comparable effects, with zeroing achieving better average performance across tasks. In contrast, mean replacement consistently leads to inferior results.

This observation aligns with our earlier findings in Section 3.2, where both scaling and zeroing exhibited similar layer sensitivity patterns. While subtle differences may arise in specific contexts, which depend on model architecture or task nature.

In practice, the choice between scaling and zeroing can depend on task-specific behavior or implementation simplicity. Both support effective plug-and-play adaptation. TaLo's strength appears to

lie not in the intervention itself, but in the strategic selection of where and when to apply it. It is the layer not the operation seems to be the more decisive factor.

| Intervention | MMBench | | | | | Avg |
|---|---|---|---|---|---|---|
| | Physical property reasoning | Structuralized i-t understanding | Attribute recognition | Celebrity recognition | Image emotion | |
| Zeroing | 55.3 ↑7.8 | 55.8 ↑0.8 | 65.2 ↓3.1 | 68.5 ↓1.4 | 65.6 ↑0.6 | 0.94↑ |
| Uniform Scaling | 57.0 ↑5.0 | 55.8 ↑0.8 | 71.0 ↑0.5 | 67.7 ↓2.8 | 65.6 ↓0.6 | 0.58↑ |

Table 8: Results of TaLo on *LLaVA-Next-LLaMA3-8B* under different intervention methods (*Structuralized i-t understanding* stands for Structuralized image text understanding).

**More Experimental Results of TaLo**    As evidenced by the comprehensive results in Table 9 10, which encompasses evaluations on MMBench and ScienceQA, TaLo demonstrates a robust ability to enhance performance across a wide spectrum of tasks. Nonetheless, the magnitude of improvement is observed to be more constrained and in some cases even decreases for particularly challenging categories, such as those involving complex multi-step reasoning, detailed visual attribute discrimination. This pattern suggests that while our proposed layer-level intervention provides an effective mechanism for task adaptation, its efficacy is bounded by the underlying capabilities of the pretrained model. Performance plateaus or regressions in these demanding scenarios likely point to limitations that are architectural or data-based in nature, which might be addressed in future work by integrating stronger inductive biases or auxiliary knowledge sources.

| Model | Shots | MMBench | | | ScienceQA | | | |
|---|---|---|---|---|---|---|---|---|
| | | F-P | OCR | F-R | Ecological interactions | The Americas: Geography | Oceania: Geography | Geography |
| | 10 shots | 43.3 ↓3.3 | 69.8 ↓2.6 | 68.6 −0.0 | 17.6 −0.0 | 30.0 ↑20.0 | 21.9 −0.0 | 41.7 ↑8.4 |
| LLaVA | 15 shots | 48.3 ↓1.1 | 71.2 −0.0 | 68.0 ↓1.3 | 29.4 ↑11.8 | 25.0 ↑5.0 | 25.0 ↑3.1 | 39.6 ↑4.2 |
| | 20 shots | 58.6 ↑6.9 | 78.8 ↑3.8 | 71.0 ↑2.2 | 29.4 ↑11.8 | 35.0 ↑20.0 | 15.6 −0.0 | 33.3 ↑2.1 |

Table 9: Additional results of TaLo on *LLaVA-Next-LLaMA3-8B*. Here, *future prediction* is annotated as F-P, and *function reasoning* is annotated as F-R.

| Model | Shots | MMBench | | | | ScienceQA | | | | |
|---|---|---|---|---|---|---|---|---|---|---|
| | | S-I-U | AR | PR | CR | Astronomy | The Americas: Geography | Genes to traits | Solutions | Force and motion |
| | 10 shots | 52.5 ↑3.3 | 72.3 ↑0.9 | 47.6 ↑1.6 | 75.0 ↑2.0 | 35.5 ↑3.2 | 25.0 ↑15.0 | 21.9 ↑12.5 | 24.4 ↑2.2 | 35.3 ↑5.9 |
| Qwen-VL | 15 shots | 52.3 ↑3.2 | 72.1 ↑2.0 | 50.8 ↑3.2 | 75.3 ↑1.5 | 38.7 ↑6.4 | 10.0 −0.0 | 25.0 ↑3.1 | 28.9 ↑8.9 | 29.4 ↓5.9 |
| | 20shots | 48.5 ↑2.5 | 73.4 ↑1.1 | 55.6 ↑3.2 | 76.6 ↑1.0 | 32.3 ↑6.5 | 15.0 ↑5.0 | 28.1 ↑6.2 | 35.6 ↑15.6 | 52.9 ↑17.6 |

Table 10: Additional results of TaLo on *Qwen2-VL-2B*, where *structuralized imagetext understanding* is annotated as S-I-U, *attribute recognition* is annotated as AR, *physical relation* is annotated as PR, and *celebrity recognition* is annotated as CR.

## C.3    EMPIRICAL VALIDATIONS OF TASK-INTERFERING LAYERS

The accuracy change heatmaps across multiple models (*LLaVA-Next-Llama3-8B*, *Qwen2-VL-2B*, and *InternVL2-40B*) under different intervention strategies are shown in Figures 7 to 18. Evaluated on diverse benchmarks, these heatmaps reveal consistent patterns of layer-specific performance gains, forming the core empirical basis for the task-interfering layers phenomenon. Rather than isolated anomalies, the results suggest a systemic trade-off in how individual layers support competing task demands, observable across model scales and architectures.

| Cluster Category and Tasks Included | |
|---|---|
| **Cluster 1 (Quantitative Reasoning)** | numeric commonsense, arithmetic reasoning, geometry reasoning, algebraic reasoning, geometry problem solving, math word problem, figure question answering, statistical reasoning, Cities, Informational texts: level 1, Particle motion and energy |
| **Cluster 2 (Analytical Reasoning)** | image_emotion, biology, engineering, public health, instance reasoning, math, geography, visual reasoning, architecture & engineering, diagnostics & laboratory medicine, electronics, psychology, maps, magnets, plant reproduction, domain-specific vocabulary, genes to traits |
| **Cluster 3 (Scientific Knowledge)** | scientific reasoning, textbook question answering, chemistry, medicine, economics, physics, sociology, art & design, science & technology, astronomy, plants, weather, fossils, thermal energy, natural resources, pharmacy, humanities & social science, literature |
| **Cluster 4 (Integrative Reasoning)** | logical reasoning, accounting, history, pharmacy, engineering practices, ecology, world religions, persuasive strategies, adaptations and natural selection, economics, sociology, humanities, design, literature, natural science theory, political history |
| **Cluster 5 (Perceptual Categorization)** | action recognition, attribute recognition, image quality, coarse perception, fine-grained perception, object localization, classification, ecosystems, force and motion, solutions, states of matter, scene understanding, OCR, image_scene, visual elements, structuralized image-text understanding, attribute comparison, celebrity recognition |
| **Cluster 6 (Predictive Reasoning)** | future prediction, identity reasoning, spatial relationship, electronics, psychology, math, music, plant reproduction, velocity & acceleration, instance interaction, physical geography, classification and scientific names, context clues, text understanding |
| **Cluster 7 (Relational Understanding)** | nature_relation, physical_relation, social relation, art theory, civil war and reconstruction, age of exploration, ancient Mesopotamia, plate tectonics, geology, animals, agriculture, ecosystems, cultural history, world religions, state capitals |

Table 11: Clusters and their included tasks from various benchmarks.

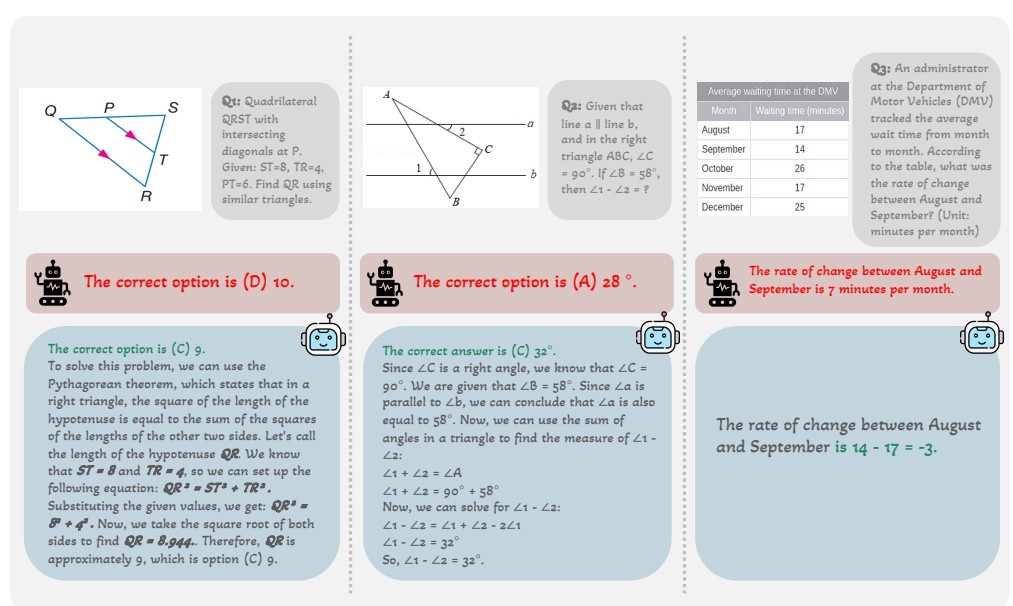

Figure 6: Qualitative Case Studies Illustrating the Effects of Layer Zeroing on LLaVA-Next's Reasoning. The figure presents three comparative examples of the model's reasoning process before (base model) and after the intervention of a specific task-interfering layer.

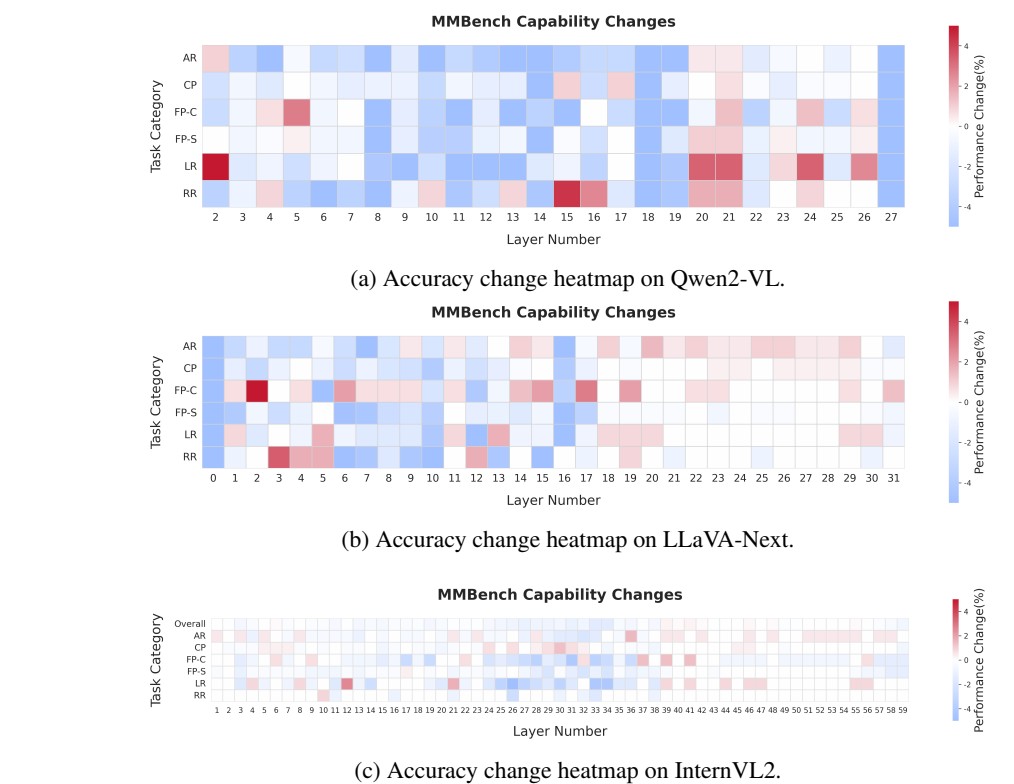

(a) Accuracy change heatmap on Qwen2-VL.

(b) Accuracy change heatmap on LLaVA-Next.

(c) Accuracy change heatmap on InternVL2.

Figure 7: Accuracy change heatmaps on MMBench (Uniform Scaling).

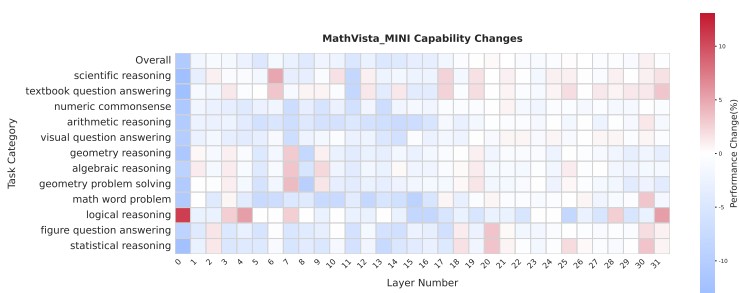

Figure 8: Accuracy change heatmap for LLaVA-Next on MathVista-MINI (Uniform Scaling).

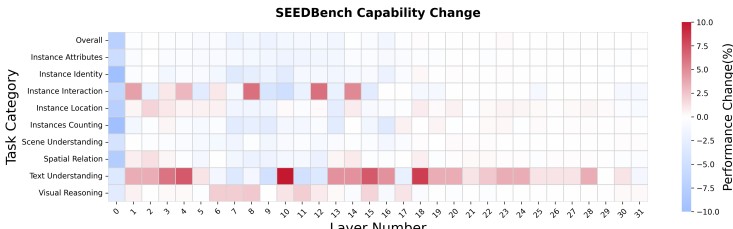

Figure 9: Accuracy change heatmap for LLaVA-Next on SEEDBench (Uniform Scaling).

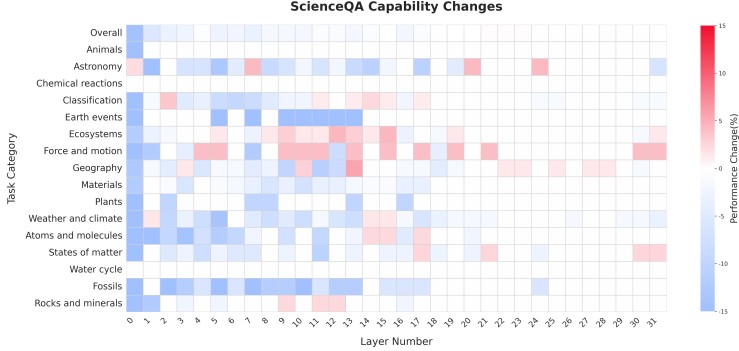

Figure 10: Accuracy change heatmap for LLaVA-Next on ScienceQA (Uniform Scaling).

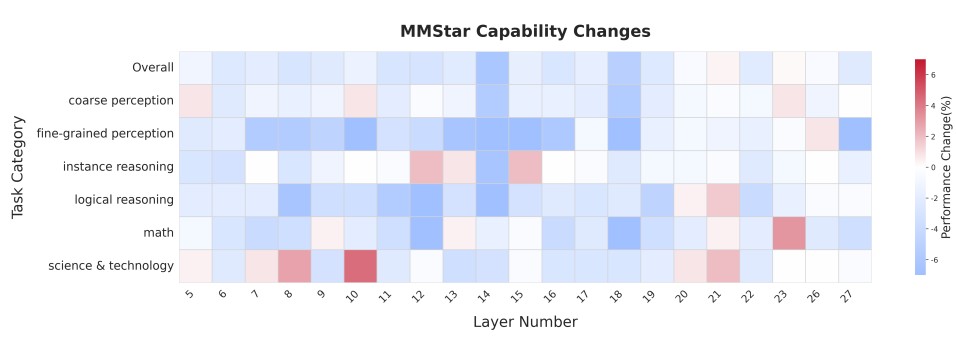

(a) Accuracy change heatmap on Qwen2-VL.

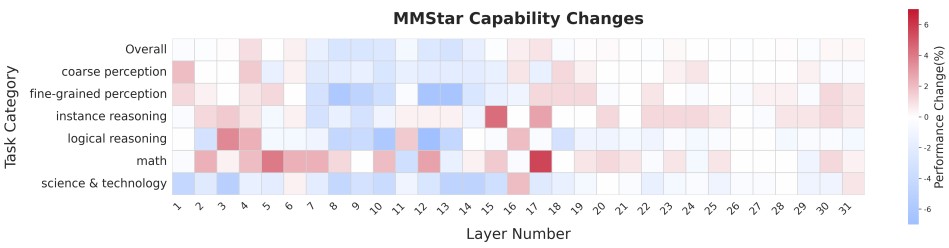

(b) Accuracy change heatmap on LLaVA-Next.

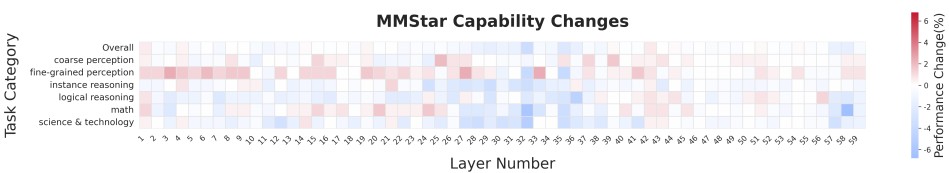

(c) Accuracy change heatmap on InternVL2.

Figure 11: Accuracy change heatmaps on MMStar (Uniform Scaling).

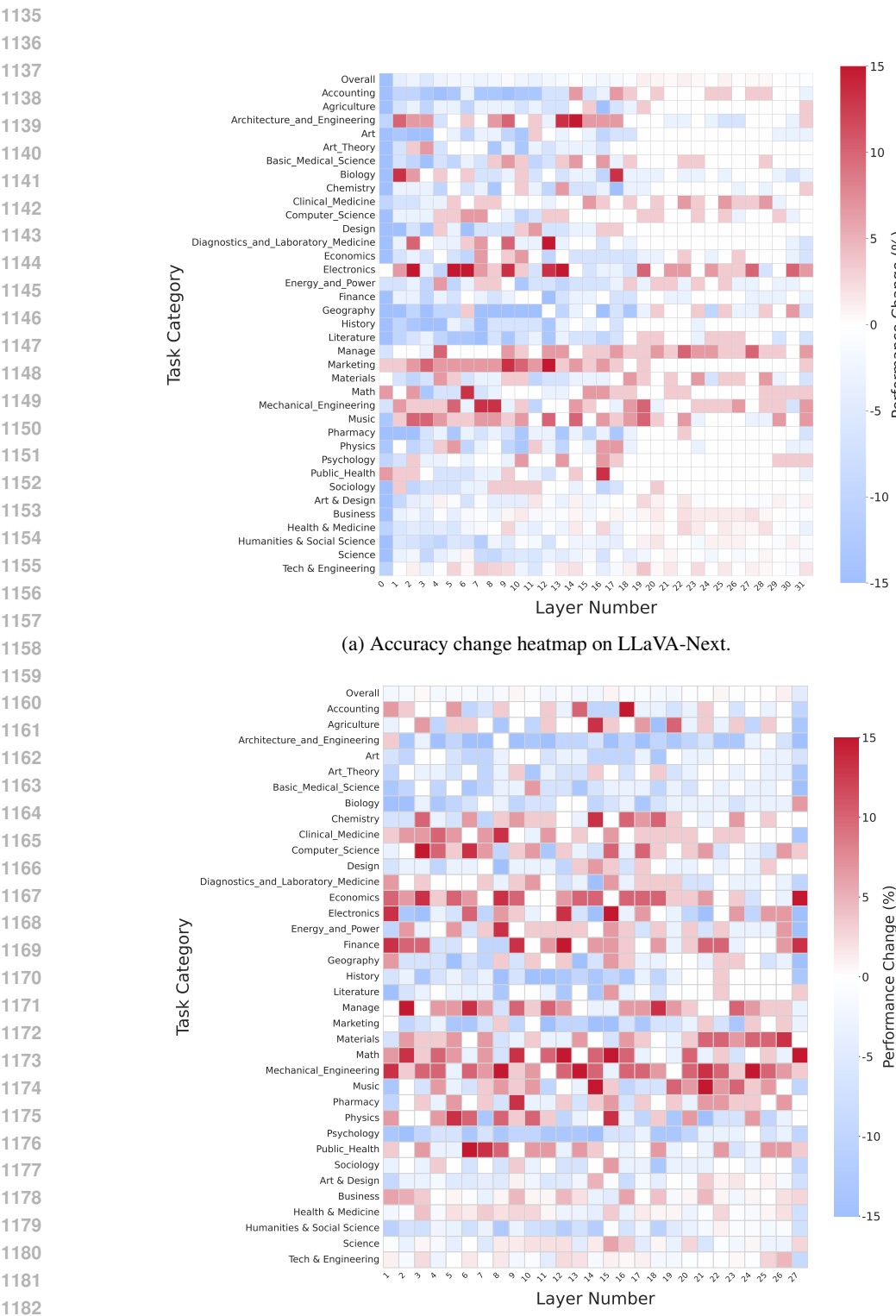

(a) Accuracy change heatmap on LLaVA-Next.

(b) Accuracy change heatmap on Qwen2-VL.

Figure 12: Accuracy change heatmaps on MMMU (Uniform Scaling).

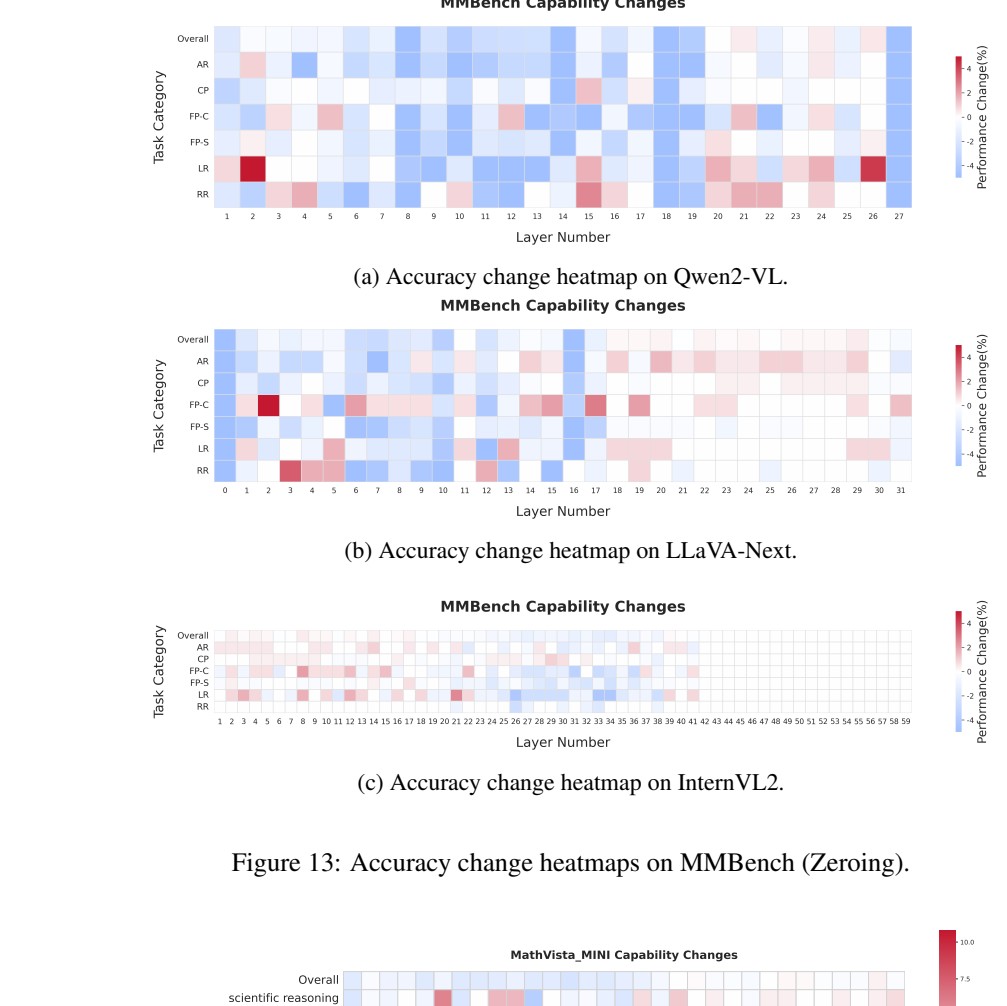

(a) Accuracy change heatmap on Qwen2-VL.

(b) Accuracy change heatmap on LLaVA-Next.

(c) Accuracy change heatmap on InternVL2.

Figure 13: Accuracy change heatmaps on MMBench (Zeroing).

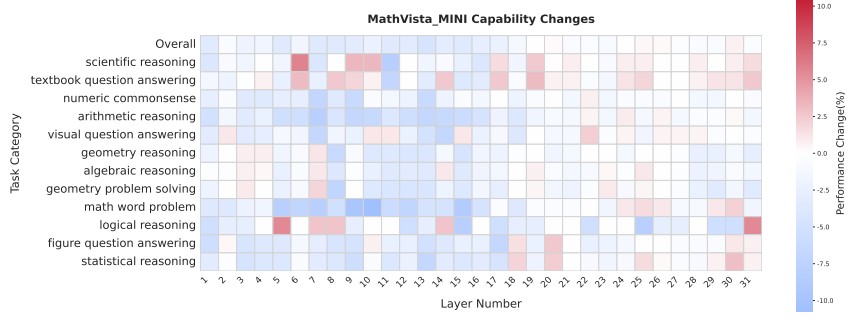

Figure 14: Accuracy change heatmap for LLaVA-Next on MathVista-MINI (Zeroing).

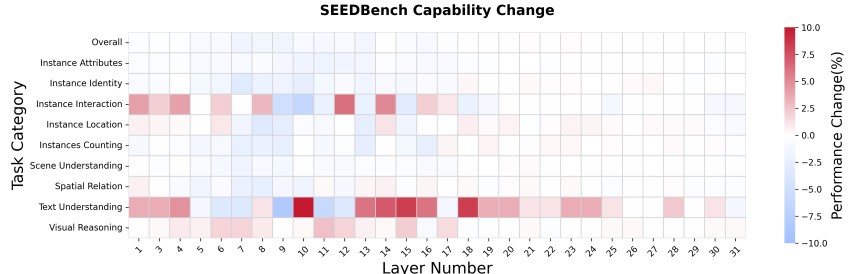

Figure 15: Accuracy change heatmap for LLaVA-Next on SEEDBench (Zeroing).

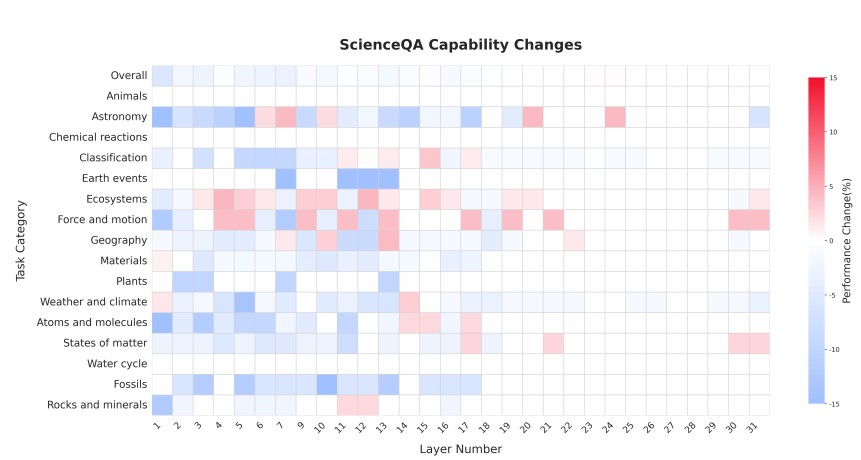

Figure 16: Accuracy change heatmap for LLaVA-Next on ScienceQA (Zeroing).

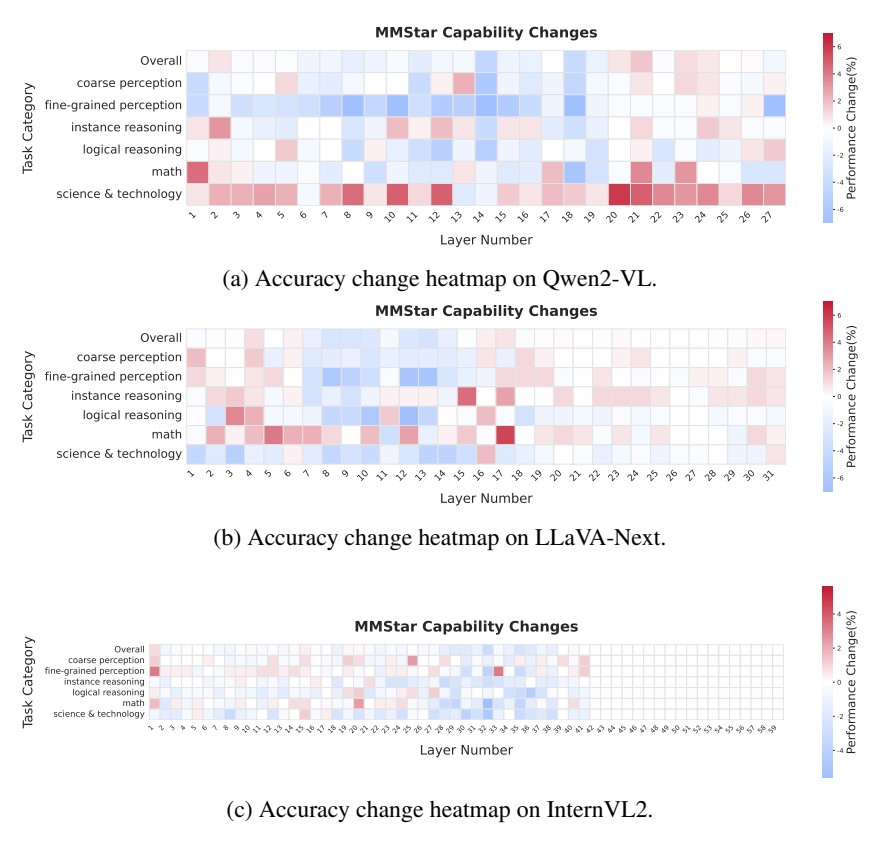

(a) Accuracy change heatmap on Qwen2-VL.

(b) Accuracy change heatmap on LLaVA-Next.

(c) Accuracy change heatmap on InternVL2.

Figure 17: Accuracy change heatmaps on MMStar (Zeroing).

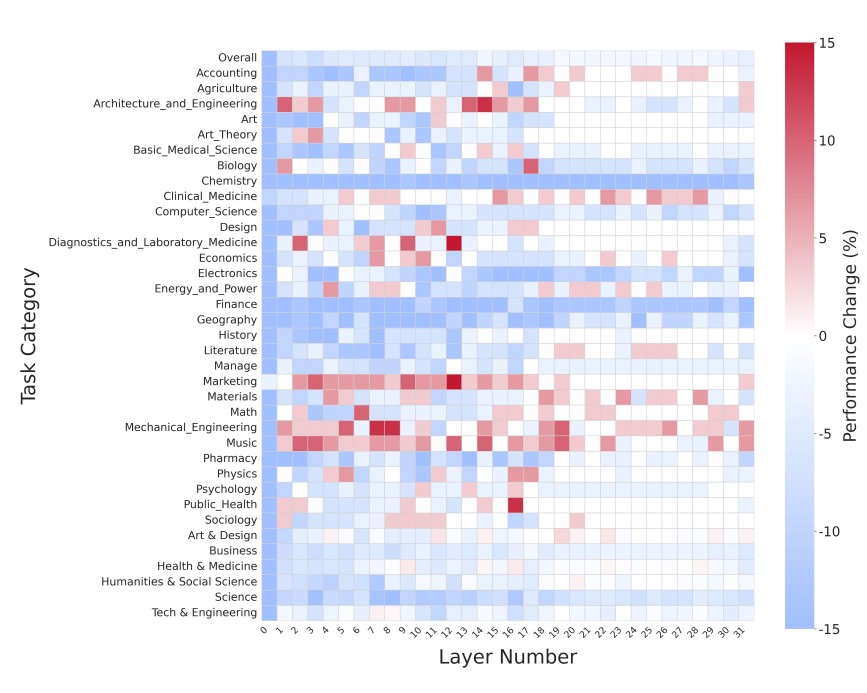

(a) Accuracy change heatmap on LLaVA-Next.

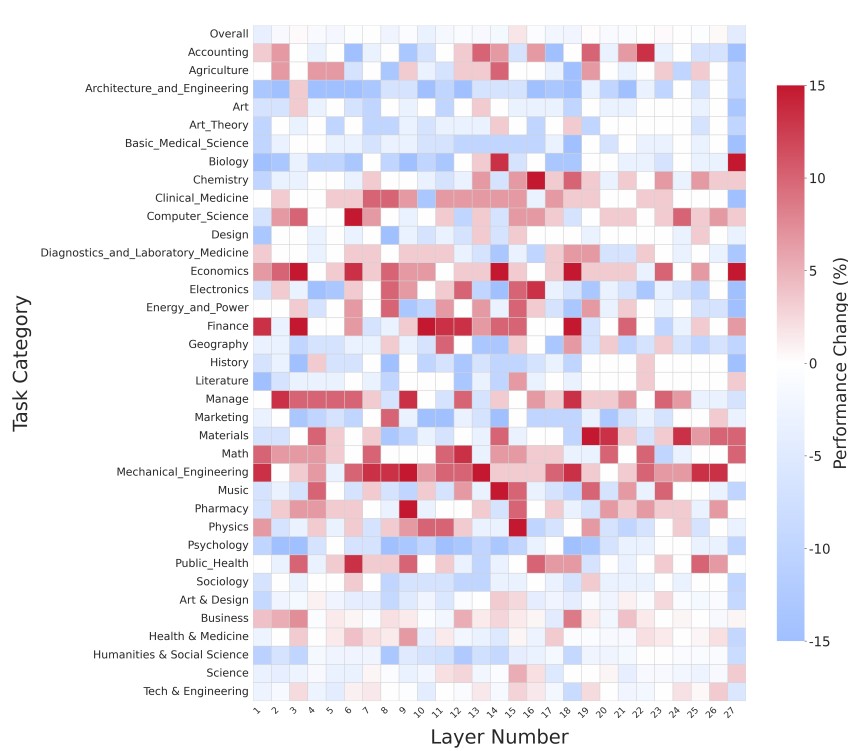

(b) Accuracy change heatmap on Qwen2-VL.

Figure 18: Accuracy change heatmaps on MMMU (Zeroing).

