# OpenReview forum: "Do All Individual Layers Help? An Empirical Study of Task-Interfering Layers in Vision-Language Models"
_ICLR.cc/2026/Conference — ICLR 2026 Conference Withdrawn Submission_

### Official Review · Reviewer_EJAA · 2025-10-15

**Soundness:** 3
**Presentation:** 3
**Contribution:** 3
**Rating:** 6
**Confidence:** 4

**Summary:**

This paper challenges the common assumption that all layers of a pretrained VLM are beneficial for every downstream task. Through a systematic empirical investigation, the authors discover that removing (zeroing out) certain individual layers can actually improve performance on specific tasks, suggesting that these layers may be task-interfering rather than helpful.

**Strengths:**

1. The writing and illustrations are clear, concise, and easy to follow.
2. The authors present a well-designed empirical framework, including layer-wise intervention, the introduction of the Task-Layer Interaction Vector, and systematic evaluation across multiple tasks and models.
3. The results are consistently validated across diverse VLM architectures (LLaVA, Qwen-VL, InternVL), benchmarks, and test sets, demonstrating robustness of the proposed method.

**Weaknesses:**

1. While the paper offers hypotheses about the origins of task-interfering layers, it's still difficult to understand why certain benchmarks consistently exhibit the phenomenon, while others seem to require all layers.
2. The study primarily evaluates pretrained models and does not investigate whether instructed finetuning (e.g., using task-specific training data such as extensive math datasets) alters the presence or impact of task-interfering layers. Assessing instructed finetuned models would help verify the generality of the phenomenon and clarify whether targeted training modifies or mitigates layer interference effects.

**Questions:**

1. Does the phenomenon of task-interfering layers stem from insufficient data in certain domains during pretraining? Would scaling up domain-specific training data reduce or eliminate such effects, or alter which layers interfere?
2. The paper finds that increasing probe set size (“more shots”) can sometimes degrade the performance of TaLo selection compared to 10-shot settings. What underlying factors might explain this trend? Is 10–20 shots a robust choice for probing task-layer interactions, and do larger probe sizes (>=100) offer benefits?

---

### Official Review · Reviewer_nRNe · 2025-10-27

**Soundness:** 2
**Presentation:** 3
**Contribution:** 2
**Rating:** 2
**Confidence:** 4

**Summary:**

This paper studies the functionality of specific layers in Vision Language Models (VLMs). It reveals that intervening on a single layer by zeroing out the attention mechanism or setting attention weights to uniform does not degrade performance on certain tasks but instead improves it. Such layers are named as Task-Interfering Layers. For each task, collecting the accuracy changes for all layers gives a Task-Layer Interaction Vector, and the paper shows that such vectors across tasks for each model demonstrate a clustering effect for similar tasks. Finally, the paper proposes Task-Adaptive Layer Knockout to identify such layers for a certain task on a held-out probing dataset and shows performance change on the test set.

**Strengths:**

The paper is well written and easy to follow.

The finding that knocking out certain layers can increase accuracy on certain tasks is novel.

The study of layer importance has been prevalent in LLMs, and the application to VLMs is new.

**Weaknesses:**

I spot the following weaknesses.

Imprecise description of “layer intervention”. The intervention occurs on the attention part, and the MLPs are never altered. In this case, the wording of “parameter zeroing” for “individual layers” is a bit misleading, and the paper has never ablated the effects of MLPs. Do they have similar effects? Will the claim still hold for MLPs? If not, the theme of the paper should be adjusted to put an emphasis on the importance of the attention mechanism, not on the full layer.

Lack of ablation. (1) For VLMs, the vision backbone and the language module are usually separated, and according to the paper, the intervention takes place on the language part. However, this choice has not been explicitly discussed, and the paper never tries to approach the vision backbone. It is perfectly fine to only study the effect of the language module, but the choice needs to be made clear with a proper explanation, plus reasonable ablation studies. (2) How do the authors pick the subsets in certain benchmarks in Table 1?

The method's applicability is constrained. For example, ⅔ out of the models tested in Tables 1 and 2 (Qwen and InternVL2), about half of the entries either show no improvement (indicating a failure to locate a specific layer in the TaLo identification phase) or show a degrading performance. This questions the effectiveness of the proposal in reality, since with high probability, users cannot obtain practical gains through such interventions.

Problematic in fine-tuning experiments in Table 3. The paper mentioned “we use … the same few-shot samples for both TaLo and the fine-tuning approaches.” For VLM fine-tuning, using a tiny set of few-shot samples does not make any practical sense, and thus makes the comparison in Table 3 not meaningful.

**Questions:**

Baselines seem to be lower than official reports. For example, with Qwen-VL on MMMU, the official report gives 41.1 (https://huggingface.co/Qwen/Qwen2-VL-2B-Instruct), and the reported numbers for the two categories are ~28 and ~34, respectively. The number from the report on the whole MMMU dataset is not directly comparable to the results in the paper, but still, do the authors use the correct chat template during evaluation? If not, the eval results in the main tables would be incorrect. Also, why do the authors choose those specific categories for testing? What happens with the remaining sets that are discarded?

---

### Official Review · Reviewer_R7SK · 2025-10-27

**Soundness:** 2
**Presentation:** 3
**Contribution:** 2
**Rating:** 4
**Confidence:** 4

**Summary:**

This paper investigates the phenomenon that selectively bypassing individual layers in a VLM can, counterintuitively, improve performance on specific downstream tasks. The authors term these layers “Task-Interfering Layers”. To systematically study this, they introduce the “Task-Layer Interaction Vector”, which quantifies the performance change for a given task when each layer is intervened upon. Their clustering analysis suggests that tasks requiring similar cognitive abilities exhibit similar interaction vectors. Based on these findings, the paper proposes TaLo (”Task-Adaptive Layer Knockout”), which uses a small “probing set” of examples to identify and bypass the most detrimental layer for a given task. The authors demonstrate that TaLo can achieve performance gains on various VLMs and benchmarks without any parameter updates.

**Strengths:**

S1. The idea and empirical observations are very interesting, and the motivation of this paper is clear.

S2. The approach is simple. It is easy to understand and to implement.

S3. The “Task-Adaptive Layer Knockout” method is effective for some tasks and computationally cheap during test-time.

S4. Their backpropagation-free “Dynamic Layer Selection” stage is lightweight computationally, which is a great advantage over fine-tuning methods. It is practical for large models under resource-constrained settings.

S5. The writing and figures are good and easy to follow.

**Weaknesses:**

W1. Limited novelty. The empirical novelty is limited, overlapping substantially with existing literature on layer importance, redundancy, and pruning. The core analysis tool, the "Task-Layer Interaction Vector," is very similar to the layer importance analysis presented in prior work in [1]. Furthermore, the intervention techniques employed are not novel. Parameter zeroing is a standard technique extensively explored in the model pruning literature. Similarly, the uniform weight intervention strategy has been utilized previously in [2].

W2. Selective reporting and cherry-picking of results. The experimental validation appears selective. The evaluation is limited to 5 MCQ benchmarks and 1 VQA benchmark, with no open-ended generation tasks. More importantly, the paper claims to analyze “nearly 100 tasks,” but the main results tables (Tables 1-3) report performance on a small, cherry-picked subset of sub-categories. This selective reporting is concerning. Given the large number of tasks tested, many of which are multiple-choice with a low random-chance baseline (e.g., 25%), it is statistically probable that some sub-categories would show performance gains purely due to randomness. The paper fails to control for this. Furthermore, the paper does not report average performance across entire benchmarks, only on the selected sub-tasks, making it difficult to assess the method's true overall impact. This selective reporting, combined with marginal average gains, suggests the practical impact is overstated and may not be generalizable.

W3. Insufficient ablations. The paper fails to justify its core design decisions with necessary ablations.

W3.1 Component choice. The intervention is limited to self-attention modules without any justification or ablation. It is unclear why FFN layers [1], widely believed to store factual knowledge, were ignored, or why whole modules were targeted instead of finer-grained components (e.g., heads, neurons [3]).

W3.2 Intervention granularity. The paper operates at a coarse, static task-level but provides no ablation for this choice. It fails to explore or compare against more dynamic instance-level or token-level adaptive strategies, which could be more powerful. The paper omits comparisons to a large body of relevant work on dynamic layer skipping [4]. These adaptive methods, which may not require a labeled calibration set for each task, represent critical baselines for the proposed task-level approach.

[1] Yang Zhang, Yanfei Dong, and Kenji Kawaguchi. 2024. Investigating Layer Importance in Large Language Models. In Proceedings of the 7th BlackboxNLP Workshop: Analyzing and Interpreting Neural Networks for NLP.

[2] Shiqi Chen, Jinghan Zhang, Tongyao Zhu, Wei Liu, Siyang Gao, Miao Xiong, Manling Li, and Junxian He. 2025. Bring Reason to Vision: Understanding Perception and Reasoning through Model Merging. In Proceedings of the Forty-second International Conference on Machine Learning (ICML 2025).

[3] Ameen Ali Ali, Shahar Katz, Lior Wolf, and Ivan Titov. 2025. Detecting and Pruning Prominent but Detrimental Neurons in Large Language Models. In Proceedings of the Second Conference on Language Modeling.

[4] Haoyu Wang, Yaqing Wang, Tianci Liu, Tuo Zhao, and Jing Gao. 2023. HadSkip: Homotopic and Adaptive Layer Skipping of Pre-trained Language Models for Efficient Inference. In Findings of the Association for Computational Linguistics: EMNLP 2023.

**Questions:**

Q1. The comparison in Table 3 shows TaLo outperforming fine-tuning with 10-20 shots. Intuitively, fine-tuning should scale better with more data (e.g., N>1000). Have you investigated this “breaking point” where the one-time cost and better scaling of fine-tuning would surpass TaLo's few-shot calibration approach?

Q2. TaLo's performance seems critically dependent on the N-shot “probing set,” yet no sensitivity analysis is provided. How robust is the selection of the “task-interfering layer”? Specifically, what is the variance in the selected layer's index if you were to repeatedly resample the 10-shot probing sets for the same task?

Q3. The Task-Layer Interaction Vectors are presented in the supplement (e.g., Figures 17, 18) but their cross-model generalization is not discussed. The vectors appear visually very different across models for the same tasks. How do you explain this? Does this imply the “interference” is more a model-specific artifact than a task-inherent property? Is it possible to develop a model-agnostic interference method for better cross-model generalizability?

Q4. The paper's explanation for why layers interfere, the “global vs. local optimum” hypothesis, is presented as a conjecture. Beyond this speculation, is there any supporting work or mechanistic interpretability analysis that provides a more grounded, causal explanation for what these layers do to cause interference, rather than just observing the correlation with performance?

---

### Official Review · Reviewer_16dw · 2025-10-30

**Soundness:** 3
**Presentation:** 3
**Contribution:** 2
**Rating:** 4
**Confidence:** 4

**Summary:**

The paper reveals that some layers in pretrained Vision-Language Models (VLMs) hinder rather than help downstream task performance, a phenomenon quantitatively validated by the proposed Task-Layer Interaction Vector. Based on this discovery, the authors introduce TaLo, a training-free test-time adaptation framework that dynamically identifies and bypasses these interfering layers, leading to consistent performance gains without retraining.

**Strengths:**

1) The proposed TaLo framework is training-free and lightweight, aligning with the current direction of test-time adaptation for Vision-Language Models. It can make a certain and meaningful contribution to this area and provide useful inspiration and reference for future studies.

2) The experimental design is rigorous and comprehensive: Across five multimodal benchmarks (MMStar, MMBench, MMMU, ScienceQA, SEEDBench), TaLo achieves consistent gains—up to +10.4% on LLaVA and +16.6% on Qwen-VL—demonstrating strong generality across models and tasks. Ablations confirm robustness: parameter zeroing and uniform scaling yield correlated effects (r > 0.95), while random-noise interventions collapse performance, validating causal layer roles. Compared with LoRA, OFT, and Merge, TaLo attains +2–4% higher accuracy with >10× faster adaptation. These extensive, multi-model experiments substantiate both the existence of Task-Interfering Layers and the efficacy of TaLo’s training-free design.

3) The paper is clearly written and well-structured, with comprehensive visualizations and quantitative analyses that effectively support the main findings.

**Weaknesses:**

1) The paper explores the internal structure of vision-language models at a coarse, layer-wise granularity, investigating how entire layers influence downstream task performance. However, it does not articulate the advantages of this approach over prior studies that conduct finer-grained, parameter-level analyses within each layer (e.g., Regularized Mask Tuning [1]). In fact, the latter may provide a more precise and informative perspective, as parameters within the same layer can simultaneously exert both facilitative and inhibitory effects on downstream tasks. Consequently, analyzing model behavior solely at the layer level may be overly coarse and somewhat restrictive in capturing the nuanced internal dynamics of vision-language models.
[1] Zheng K, Wu W, Feng R, et al. Regularized mask tuning: Uncovering hidden knowledge in pre-trained vision-language models[C]//Proceedings of the IEEE/CVF International Conference on Computer Vision. 2023: 11663-11673.

2) The paper does not specify the criteria or statistical thresholds used to determine when a layer qualifies as task-interfering. Although the metric ∆ℓ (accuracy gain) is formally defined, the lack of significance testing or confidence evaluation makes it difficult to assess the statistical robustness and reliability of the identified task-interfering layers.

3) As acknowledged by the authors in the Limitation and Future Work section, the research idea and framework design of the paper are somewhat simplistic. In my view, it serves primarily as a proof of concept rather than a fully developed methodological contribution.

**Questions:**

1) Could the authors elaborate on the rationale for adopting a layer-wise analysis rather than a parameter-level one? What concrete advantages—such as improved interpretability, computational efficiency, or generalizability—does the layer-level perspective offer? Additionally, since parameters within a single layer may exert both positive and negative effects on downstream performance, how might this internal heterogeneity affect the reliability and interpretability of the proposed approach?

2) What quantitative or statistical criteria are applied to determine when a layer qualifies as task-interfering? Have the authors considered incorporating methods such as significance testing, bootstrapping, or confidence interval estimation to better assess the robustness and statistical validity of ∆ℓ across tasks and datasets?

3) Given the simplicity of the proposed framework, how do the authors plan to further develop it into a more comprehensive or theoretically grounded methodology? For example, could future work incorporate multi-layer interaction modeling, adaptive layer selection mechanisms, or task-conditioned weighting strategies to improve the method’s scalability, generality, and theoretical depth?

---

### Note · Authors · 2025-11-13

I have read and agree with the venue's withdrawal policy on behalf of myself and my co-authors.